# Alternatively Spliced Isoforms of *MUC4* and *ADAM12* as Biomarkers for Colorectal Cancer Metastasis

**DOI:** 10.3390/jpm13010135

**Published:** 2023-01-10

**Authors:** Saleh Althenayyan, Mohammed H. AlMuhanna, Abdulkareem AlAbdulrahman, Bandar Alghanem, Suliman A. Alsagaby, Abdulaziz Alfahed, Glowi Alasiri, Mohammad Azhar Aziz

**Affiliations:** 1Colorectal Cancer Research Program, Department of Cellular Therapy and Cancer Research, King Abdullah International Medical Research Center, Riyadh 11481, Saudi Arabia; 2King Saud Bin Abdulaziz University for Health Sciences, Riyadh 11481, Saudi Arabia; 3Department of Medical Genomics, King Abdullah International Medical Research Center, Riyadh 11481, Saudi Arabia; 4Department of Core Facilities and Platforms, King Abdullah International Medical Research Center, Riyadh 11481, Saudi Arabia; 5Department of Medical Laboratory Sciences, College of Applied Medical Sciences, Majmaah University, Majmaah 11932, Saudi Arabia; 6Department of Medical Laboratory Sciences, College of Applied Medical Sciences, Prince Sattam bin Abdulaziz University, Alkharj 11942, Saudi Arabia; 7Department of Biochemistry, College of Medicine, Al Imam Mohammad Ibn Saud Islamic University (IMSIU), Riyadh 13317, Saudi Arabia; 8Interdisciplinary Nanotechnology Center, Aligarh Muslim University, Aligarh 202002, India

**Keywords:** colorectal cancer, alternative splicing, mucins, biomarkers, precision medicine, preventive diagnostics, prevention, personalized medicine

## Abstract

There is a pertinent need to develop prognostic biomarkers for practicing predictive, preventive and personalized medicine (PPPM) in colorectal cancer metastasis. The analysis of isoform expression data governed by alternative splicing provides a high-resolution picture of mRNAs in a defined condition. This information would not be available by studying gene expression changes alone. Hence, we utilized our prior data from an exon microarray and found *ADAM12* and *MUC4* to be strong biomarker candidates based on their alternative splicing scores and pattern. In this study, we characterized their isoform expression in a cell line model of metastatic colorectal cancer (SW480 & SW620). These two genes were found to be good prognostic indicators in two cohorts from The Cancer Genome Atlas database. We studied their exon structure using sequence information in the NCBI and ENSEMBL genome databases to amplify and validate six isoforms each for the *ADAM12* and *MUC4* genes. The differential expression of these isoforms was observed between normal, primary and metastatic colorectal cancer cell lines. RNA-Seq analysis further proved the differential expression of the gene isoforms. The isoforms of *MUC4* and *ADAM12* were found to change expression levels in response to 5-Fluorouracil (5-FU) treatment in a dose-, time- and cell line-dependent manner. Furthermore, we successfully detected the protein isoforms of *ADAM12* and *MUC4* in cell lysates, reflecting the differential expression at the protein level. The change in the mRNA and protein expression of *MUC4* and *ADAM12* in primary and metastatic cells and in response to 5-FU qualifies them to be studied as potential biomarkers. This comprehensive study underscores the importance of studying alternatively spliced isoforms and their potential use as prognostic and/or predictive biomarkers in the PPPM approach towards cancer.

## 1. Introduction

Colorectal cancer (CRC) is the third most common type of cancer in the world [1,2] and the second-leading cause of death in cancer patients [3]. The clinical outcome of CRC is unsatisfactory, especially in patients with the advanced metastatic stages of the disease, where the 5-year survival rate does not exceed 10% [4,5]. Different factors, such as family history, a poor lifestyle, inflammatory bowel diseases, diabetes mellitus and cholecystectomy, have been reported to increase the risk of developing CRC [6]. Defects in the chromosomal instability (CIN) pathway (in particular, mutations in the *APC*, *TP53* and *KRAS* genes), microsatellite instability (MSI), mutations in the *BRAF* gene and the loss of function in *P16* were implicated in the initiation and progression of CRC [3,7,8]. As a complex disease with molecular heterogeneity, CRC patients were reported to fall into four subgroups with distinct (i) molecular bases, (ii) prognoses (iii) and (iv) responses to treatments [9]. Several studies have provided insights into the molecular mechanisms through which the disease initiates and progresses and proposed methods for patient stratification, the prediction of clinical outcomes and personalized medicine. Personalized medicine has attracted interest for decades and, with the aid of biomarkers, is now complementing therapeutic decision-making processes in clinical settings.

Biomarkers are statistically significant, measurable indicators of a disease. Along with the traditional staging system and microsatellite instability status, prognostic biomarkers are used to predict the progression of a disease, and predictive biomarkers can be used to predict the efficacy of a particular therapeutic regimen. Although the microsatellite instability status and the location of the tumor (right- or left-sided) can help to guide treatment decision making in CRC, there is still a lack of accurate prognostic and predictive molecular biomarkers for metastatic CRC [10,11]. Thus, there is an unfulfilled need to develop molecular biomarkers that can help to predict the course of disease progression after administering the standard-of-care regimens for metastatic CRC.

Isoforms of a gene may serve as precise biomarkers. A gene isoform is defined as an alternatively spliced form of messenger RNA. Alternative splicing (AS)-based biomarkers have not been studied in detail due to their technical complexity but may hold significant potential as biomarkers in complex diseases such as cancer and diabetes. Altered splicing machinery has been linked to cancer [12]. More than 90% of genes are alternatively spliced, and disturbances to this complex machinery could result in the initiation and progression of cancer [13,14] Molecular changes in splicing genes and altered ratios of abundance of splice variants have been associated with the occurrence, increased progression and adverse prognosis of many types of cancer, including colorectal, breast, prostate and blood cancer [15,16,17,18]). Therefore, studying the alternative splicing (AS) events happening in CRC would allow for the identification of biomarkers that could be used for predictive diagnosis, prognosis, patient stratification and treatment selection.

The notion of 3PM (predictive, preventive and personalized medicine) holds great promise for providing the needed healthcare for patients with a complex disease such as cancer [19]. In cancer, the great amount of data generated by omics approaches, such as genomics [20], transcriptomics [21], proteomics [22] and metabolomics [23], improved our understanding of the underlying molecular causes of cancer and identified prognostic markers for the prediction of cancer behavior, therapeutic targets for drug development and actionable biomarkers for therapy selection [24]. Collectively, this knowledge enabled the early diagnosis of cancer, the accurate monitoring and prediction of the cancer clinical course and the selection of personalized therapy [25,26]. Despite the efforts put into CRC research, it remains a disease with poor clinical outcomes, especially in the metastatic stage, necessitating an evidence-supported paradigm shift towards the optimal application of 3PM.

Given the proven significance of AS in cancer and the need for the identification of predictive biomarkers in CRC, we characterized the AS isoforms of *MUC4* and *ADAM12* in CRC. From our previous exon profiling study [27], we selected *MUC4* and *ADAM12* for further analysis, as they had the highest, most significant exon-level expression changes. In patients with CRC, alterations of these two genes were reported at multiple levels, including the cytogenetic, gene expression and splicing levels [27,28]. In the present work, we assessed the prognostic significance of these two genes in CRC using The Cancer Genome Atlas (TCGA) database, which was preceded by confirming their differential expression in our patient cohort. Further, we compared the expression of isoforms of the *MUC4* and *ADAM12* genes in a non-cancerous intestinal epithelial cell line (CCD841) and primary (SW480) and metastatic (SW620) CRC cell lines. Sanger sequencing of specific alternatively spliced isoforms was conducted to confirm their identity. A comparison analysis of the relative abundance of the gene transcripts and proteins in the cell lines (CCD841, SW480 and SW620) was also performed. Global gene expression profiling (RNA-Seq) was employed to interrogate the expression of the *MUC4* and *ADAM12* gene isoforms in two CRC cell lines (SW480 and SW620). RNA-Seq data also provided an avenue for studying the expression changes in the pathways and molecules associated with AS. This comprehensive analysis of *MUC4* and *ADAM12* may help in the development of specific isoforms of these genes as prognostic biomarkers for CRC and contribute to the knowledge needed for the application of 3PM in this malignancy.

## 2. Materials and Methods

### 2.1. Cell Culture

CCD841, SW480 and SW620 cell lines were obtained from the American Type Culture Collection (ATCC, Manassas, VA, USA). CCD841 cells were derived from normal colonic mucosa, whereas SW480 and SW620 cells were derived from primary and metastatic CRC cells, respectively. Cells were cultured in 5% CO_2_ at 37 °C in advanced Dulbecco’s modified Eagle’s medium (DMEM), containing 10% FBS, 100 IU/mL penicillin, 100 µg/mL streptomycin and 2 mmol/L L-glutamine (Gibco, Thermo Fisher Scientific, Grand Island, NY, USA).

### 2.2. RNA Extraction and cDNA Synthesis

The total RNA was extracted from CCD841, SW480 and SW620 cells using PureLink RNA Mini Kits (Cat# 12183025, Thermo Fisher Scientific). Then, 2 µg RNA was reverse transcribed to cDNA using High-Capacity cDNA Reverse Transcription Kits (Cat# 4374967; Applied Biosystems/Thermo Fisher Scientific).

### 2.3. Primer Design

Alternatively spliced isoforms of *ADAM12* and *MUC4* were analyzed using SnapGene software (GSL Biotech, San Diego, CA, USA). A global alignment (Needleman–Wunsch) algorithm was used to align two sequences; the Clustal Omega option in Snapgene was used to align more than two sequences. The *ADAM12* and *MUC4* isoform sequences were obtained from the NCBI and ENSEMBL genome databases. Five well-characterized *ADAM12* isoforms were aligned to identify the differences between the isoforms. Three sets of primers were designed to amplify all isoforms, isoforms 1 and 3 and isoforms 2, 4 and 5 of *ADAM12*. A similar design strategy was employed to design four sets of primers to detect four well-described isoforms of *MUC4*. Both *GAPDH* and *ACTNB* were used as housekeeping control genes. The same primers were used for both endpoint PCR and qRT-PCR.

### 2.4. Semi-Quantitative Endpoint PCR

The primers described above were used to amplify alternatively spliced isoforms of *ADAM12* and *MUC4* using Master Mix DreamTaq^TM^ Hot Start Green (Cat# K9022, Thermo Fisher Scientific) on a Veriti^TM^ 96-Well Thermal Cycler (Cat# 4375786; Applied Biosystems). The PCR cycling conditions were 95 °C for 5 min, followed by 40 cycles of 95 °C for 1 min, 60 °C for 30 s and 72 °C for 30 s, followed by one cycle of 72 °C for 7 min and a 4 °C hold. All PCR products were analyzed using 2% horizontal agarose gels; a 50 bp ladder was used to estimate the band sizes.

### 2.5. Gel Extraction and Purification of Amplified PCR Products

We excised the gel slices containing DNA fragments. The gel slices were placed into 1.5 mL tubes and weighed. The binding buffer was added at a 2:1 volume ratio, and the tubes were incubated at 50–60 °C for 10 min or until the gel slices were completely dissolved. One gel volume of 100% isopropanol was added (because the band sizes were *≤*500 bp), and the DNA fragments were extracted using the Thermo Scientific GeneJET Gel Extraction Kit (cat# K0692), following the manufacturer’s protocol. The purity and quantity of the extracted DNA samples were assessed using a Nanodrop Spectrophotometer (Thermo Fisher Scientific).

### 2.6. Sanger Sequencing

The PCR products were sequenced using a DNA Analyzer (Cat# 3730XL; Applied Biosystems). The ExoSAP-IT^TM^ kit (Cat# 78200.200.UL; Thermo Fisher Scientific) was used to clean up the endpoint PCR products. Cycling sequencing was performed using the BigDye^TM^ Terminator v3.1 Kit (Cat# 4337455; Thermo Fisher Scientific). The BigDye XTerminator^TM^ Purification Kit was used to clean-up the cycling sequencing reactions (Cat# 4376486; Thermo Fisher Scientific).

### 2.7. Quantitative Real-Time PCR

Quantitative real-time PCR (qRT-PCR) was performed using either SYBR green or TaqMan Universal PCR Master Mix (Applied Biosystems; Cat# 4304437) and the QuantStudio 6 Flex Real-Time PCR System (Thermo Fisher Scientific), according to the manufacturers’ instructions. Gene expression was analyzed using TaqMan probes for *ADAM12* (Thermo Fisher Scientific; Cat# 4331182) and *MUC4* (Thermo Fisher Scientific; Cat# 4331182). Eukaryotic 18S rRNA was used as an endogenous control gene (Thermo Fisher Scientific; Cat# 4333760T). All reactions were performed in triplicate, and the qRT-PCR data were analyzed using the relative quantitative (RQ) method (2^−∆∆^*Ct*) and Expression Suite software version 1.1 (Thermo Fisher Scientific). For SYBR green PCR reactions, the same primers used for the endpoint PCR were used with SYBR^TM^ Green PCR Master Mix (Cat# 4309155; Applied Biosystems). All experiments were performed in triplicate and repeated at least twice.

### 2.8. RNA Sequencing

The total RNA was extracted using PureLink RNA Mini Kits (Cat# 12183025; Ambion). The RNA quality and concentration were evaluated using an ND-1000 UV-Vis Spectrophotometer; high-quality samples were selected for RNA-Seq. For RNA-Seq, poly (A) messenger RNA was captured using the RiboMinus^TM^ Eukaryote System v2 (Cat# 4481370), residuals were removed from the purified samples using the RiboMinus^TM^ Magnetic Bead Cleanup Module (Cat# 4481370) and the quantity of the RNA was assessed using a Qubit^TM^ RNA HS Assay kit (Cat# 032852). All procedures were performed according to the manufacturers’ protocols.

RNA libraries were constructed using the Ion Total RNA-Seq Kit v2 (Cat# 4475936), they were converted to cDNA using the Ion Total RNA-Seq Primer Set v2 (Cat# 4474810), the cDNA was indexed using the Ion Express^TM^ RNA-Seq Barcode 01-16 Kit (Cat# 4475485) and the concentrations of cDNA were measured using a NanoDrop1000. The highly qualified libraries were separately injected onto the Ion PI^TM^ Hi-Q^TM^ Chip Kit v3 (Cat# A26770) and inserted into the Ion Chef^TM^ Instrument (4484177) for emulsification and enrichment; then, sample sequencing was performed using an Ion Proton Semiconductor Sequencer (Thermo Fisher Scientific).

### 2.9. Western Blotting

Whole-cell extracts were prepared using NP40 buffer (Invitrogen, Paisley, UK) containing a protease inhibitor cocktail (Sigma-Aldrich, Taufkirchen, Germany). The protein lysates were incubated on ice for 30 min and centrifuged at 13,000 rpm for 10 min, and the supernatants were collected. Protein concentrations were determined using the Qubit Protein Assay Kit (Thermo Fisher Scientific). Then, 50 µg of whole-cell lysates was mixed with 4X Laemmli buffer (Bio-Rad), loaded onto 4–20% Mini-PROTEAN^®^ TGX^TM^ Precast Protein Gels (Bio-Rad), electrophoresed at 100 V for approximately 90 min and transferred onto polyvinyl PVDF membranes overnight at 5 V using a semidry transfer cell (Bio-Rad). The membranes were blocked in 5% bovine serum albumin (Millipore, Germany) in TBST (Tris-buffered saline Tween) for 1 h with shaking at room temperature; then, they were incubated with rabbit *MUC4* (1:1000; Cat# orb399150 & Cat# orb306041, Biorbyt) or *ADAM12* (1:1000; Cat# orb155592 & Cat# orb373867) antibodies in 5% BSA with gentle shaking overnight at 4 °C. Mouse anti-GAPDH was used as a loading control (1:10,000; Cat# orb234217). The membranes were incubated with goat anti-rabbit secondary antibodies (Cat# orb43514) for *MUC4* and *ADAM12* and goat anti-mouse antibodies for GAPDH (Cat# orb500708) at 1:10,000 at room temperature for 1 h. The signals were detected using Chemiluminescent HRP Substrate (Bio-Rad). Images were captured and analyzed using a Chemidoc gel documentation system (Bio-Rad).

### 2.10. Statistical Analysis

Prism software (GraphPad) and Microsoft Excel were employed to analyze the data and calculate *p*-values using the statistical tests indicated within the figure legends. Differences between the groups were considered significant for *p* values ≤ 0.05.

### 2.11. Survival Analysis

Two colorectal adenocarcinoma transcriptomic datasets published by The Cancer Genomic Atlas (TCGA) research network (National Cancer Institute, The Cancer Genome Atlas, NIH (Accessed 25 May 2021); Available from: https://cancergenome.nih.gov/) were used: “colorectal adenocarcinoma TCGA Firehose Legacy” and “colorectal adenocarcinoma TCGA PanCancer Atlas”. The prognostic importance of *ADAM12* and *MUC4* in these datasets was examined using “cBioPortal” coupled with Onco Query Language (OQL) tools. The selected genomic profile for the analyses in both datasets was the mRNA expression (RNA sequencing) z-scores relative to all samples (log RNA Seq V2 RSEM).

## 3. Results

### 3.1. Expression of ADAM12 and MUC4 in Human CRC

We previously identified significant alterations in the exon-level patterns of gene expression in tumor samples from our cohort of patients. Here, we further analyzed the expression patterns of isoforms of the *ADAM12* and *MUC4* genes, which were among the most significantly alternatively spliced genes in CRC. To begin with, we analyzed the expression levels of these two genes in patient tumor and normal samples: *ADAM12* was significantly upregulated, while *MUC4* was significantly downregulated in the tumor samples (Figure 1A,B).

### 3.2. Potential of ADAM12 and MUC4 as Biomarkers in CRC

The potential prognostic value of *ADAM12* and *MUC4* was assessed using two TCGA transcriptomics datasets. In the “colorectal adenocarcinoma TCGA Firehose Legacy” dataset, patients with a high expression of *ADAM12* (z-score > 0.2) had a poorer overall survival ((OS); median OS = 63 months) compared to patients with a low *ADAM12* expression (z-score < 0.2; median OS = undefined; Figure 2A; *p* = 0.006; HR = 1.84). Similarly, *ADAM12* appeared to have prognostic value for disease-specific survival (DSS) in the “colorectal adenocarcinoma TCGA PanCancer Atlas” dataset; at month 75, the proportion of DSS events was 16% (shorter DSS) in the high-*ADAM12*-expression group (z-score > 0.3), compared to 11% (longer DSS) in the low-expression group (z-score < 0.03; Figure 2B, *p* = 0.05, HR = 1.61). This finding demonstrated that a high expression of *ADAM12* predicts a shorter DSS.

In contrast, a high expression of *MUC4* (z-score > −0.83) was associated with a better OS (median OS = 100 months) compared to a low expression of *MUC4* (z-score < −0.83; median OS = 54 months; Figure 2C, *p* < 0.05, HR = 0.59). In support of this finding, *MUC4* was also associated with DSS in the “colorectal adenocarcinoma TCGA PanCancer Atlas” dataset. Figure 2D shows that the proportion of DSS events at month 75 was 8% (longer DSS) in the high-*MUC4*-expression group (z-score > 0.62), compared with 15% (shorter DSS) in the low-expression group (z-score < 0.62; *p* = 0.02, HR = 0.59). This finding indicated that a low expression of *MUC4* is predictive of a shorter DSS.

### 3.3. In Silico Identification of ADAM12 Isoforms

In order to detect *ADAM12* isoforms, we obtained the sequences for the five isoforms of *ADAM12* from the NCBI database. Table 1 provides the details of the five transcripts, along with the annotations from the ENSEMBL database and the three sets of primers we designed to amplify the groups of *ADAM12* transcripts. Isoforms 1 and 3 do not contain exon 23. *ADAM12-1* and *2* contain a nine-nucleotide sequence of (GTAATTCTG) that does not exist in *ADAM12-3*, *4* and *5*. *ADAM12-2*, *4* and *5* are the only isoforms that contain exon 23. *ADAM12-4* has a six-base pair sequence that is not unique among the isoforms but is unique at this location when all isoforms are aligned. One primer set was designed to amplify all five protein-coding isoforms of *ADAM12* (*ADAM12*-All). We considered isoforms 2, 4 and 5 (*ADAM12-245*) as one isoform for the primer design, as their sequence similarity exceeds 99%. A similar approach was employed for isoforms 1 and 3 (*ADAM12-13*) (see Figure 3A). The location of these primers and the organization of the *ADAM12* exons are illustrated in Figure 3B.

### 3.4. In Silico Identification of MUC4 Isoforms

Similarly, we used the annotations from the NCBI database to identify isoforms of *MUC4*. Appendix A provides details of the four transcripts along with the annotations from the ENSEMBL database. According to the NCBI database, *MUC4* isoform 1 is 16,756 bp, *MUC4* isoform 4 is 4048 bp, *MUC4* isoform 5 is 3895 bp and *MUC4* isoform 6 is 22,824 bp.

We designed a set of primers called *MUC4*-All to amplify all four *MUC4* protein-coding isoforms. There is a high similarity (99%) between *MUC4* isoform 1 (*MUC4-1*) and *MUC4* isoform 6 (*MUC4-6*) without their corresponding tandem repeat sequences; thus, we designed the *MUC4-16* primer set to amplify these two isoforms. There are two gap regions and seven mismatches (only two are represented in the figure) between the two sequences (Figure 4A). The skipping of exon 2 and 25 is common to all isoforms. Additionally, *MUC4-4* shows the skipping of exons 2 and 3, and *MUC4-5* shows the skipping of exons 2, 3 and 4. AS at exon junctions creates unique isoforms; thus, we employed the junctions at exons 1 and 5 to design the *MUC4-4* primers. The locations of the primers, along with the exon structure, are illustrated in Figure 4B.

### 3.5. Amplification of ADAM12 and MUC4 Isoforms

When the *ADAM12*-All, *ADAM12-245* and *ADAM12-13* primers were used, endpoint PCR resulted in specific bands of the expected sizes (Appendix A). Among these three sets, the *ADAM12*-All primers led to the lowest band intensities. The *ADAM12*-All product was differentially expressed across the three cell lines, with the highest levels in CCD841 cells, followed by SW480 cells and SW620 cells (Appendix A). No apparent differences in the band intensities of the *ADAM12-245* and *ADAM12-13* bands were observed between the three cell lines (Appendix A).

Further, we confirmed the sequences of the ADAM12-All (Appendix A), ADAM12-245 (Appendix A) and ADAM12-13 (Appendix A) amplicons in CCD841, SW480 and SW620 cells. Sanger sequencing resulted in NCBI Blast alignment confidence scores of 99 and 100%. Also the alignment gave the same results from SnapGene software.

Semi-quantitative estimations of the endpoint PCR showed that CCD841 cells had the lowest band intensity for the MUC4-All amplicon, with similar MUC4-All band intensities observed in SW480 and SW620 cells (Appendix A). Specific, single bands corresponding to the expected amplicon sizes were amplified using MUC4-1&6 and MUC4-4 primer sets (Appendix A).

The amplicons produced using the primers: MUC4-All, MUC4-16 and MUC4-4, were subjected to Sanger sequencing. We aligned the sequences with the reference isoform, MUC4-All (Appendix A), MUC4-16 (Appendix A), MUC4-4 (Appendix A) using SnapGene and blasted the sequences using the NCBI tool. The confidence scores for the MUC4-All, MUC4-16 and MUC4-4 amplicons ranged between 95 and 100%.

We were unable to detect a single, specific amplicon for MUC4-6 and MUC4-1; both the MUC4-6 (Appendix A) and MUC4-1 (Appendix A) primer sets resulted in a variety of amplicons that migrated as multiple bands. As the MUC4-1 and MUC4-6 primer sets amplified multiple bands, we excised two specific bands of the expected sizes for isoforms 1&6, as well as two other non-specific bands (Appendix A). Sanger sequencing showed that these four bands corresponded to MUC4-6 (Appendix A) and MUC4-1 (Appendix A), with the same confidence scores across all three cell lines. The confidence scores for the MUC4-1 and MUC4-6 amplicons ranged between 91% and 98%. The exception was for one band (320 kb; unexpected band size) that was amplified by the MUC4-6 primer set in SW480 cells; the confidence scores was 80% (Appendix A).

### 3.6. Quantitative Assessment of the Expression of ADAM12 and MUC4 Isoforms

The assessment of *ADAM12* expression using the *ADAM12*-All primer sets revealed *ADAM12* expression at significantly lower levels in SW480 cells than in CCD841 cells (RQ = 0.25), and the *ADAM12* expression in SW620 cells was approximately half the level observed in SW480 cells (RQ = 0.1). The isoforms detected by the *ADAM12-245* primers were expressed at similar levels in CCD841 and SW480 cells, whereas SW620 expressed almost double the levels of the *ADAM12-245* isoform (RQ = 2). The expression levels of the isoform amplified by the *ADAM12-13* primers were similar among the three cell lines (Figure 5A).

The *MUC-4*-All primers detected the highest expression of *MUC4* in SW620 cells, followed by SW480 cells. However, the *MUC4-16* primers detected the highest levels of transcripts in CCD841 cells, lower levels in SW480 cells and the lowest levels in SW620 cells. The *MUC4-4* primers resulted in a similar expression pattern as that of the *MUC4-16* primers by a decrease in expression from normal to metastatic cell lines, but no significant difference in expression values between the three cell lines was observed (Figure 5B).

### 3.7. Effect of 5-Fluorouracil on the Expression of ADAM12 and MUC4 Isoforms

We studied the effect of 5-fluorouracil (5-FU) on the expression of the isoforms of *ADAM12* and *MUC4* in SW480 and SW620 cells treated with 0, 1, 10 or 100 µM 5-FU for 24, 48, 72 or 96 h. The highest expression of the *ADAM12*-All amplicon was detected in SW620 cells treated with 100 µM 5-FU for 24 h, and the lowest level was detected in the same cells treated with 10 µM for 24 h (Figure 6A). SW480 cells treated with 5-FU for 96 h showed the highest expression of the *ADAM12*-245 amplicon, while the treatment of the same cells with 1 µM 5-FU for 24 h led to the largest downregulation of the amplicon (Figure 6B).

The highest expression of *MUC4*-All was in observed in SW620 cells treated with 100 µM 5-FU for 24 h, whereas the lowest expression was observed in SW480 cells treated with 100 µM 5-FU for 24 h (Figure 6C). The *MUC4*-4 amplicon was expressed at the highest levels in SW480 cells treated with 10 µM 5-FU for 96 h, and the lowest levels were detected in SW620 cells treated with 1 µM 5-FU for 72 h (Figure 6D).

### 3.8. RNA-Seq Isoform Analysis

We used RNA-Seq to quantify the expression of the five isoforms of *ADAM12* and the four isoforms of *MUC4* in the NCBI RefSeq database. The RNA-Seq results were annotated according to ENSEMBL GRCh37. The expression level of each *ADAM12* and *MUC4* isoform, quantified in Fragments Per Kilobase of transcript per million mapped reads (FPKM), is shown in Table 2.

We also analyzed the RNA-Seq data to explore the possible role of other AS genes in CRC. Among the six genes known to be involved in AS, *FUBP1* was found to be upregulated by >1.5-fold in SW620 cells compared to SW480 cells, whereas *U2AF1L4* was significantly downregulated (fold change 0.47; Table 3).

### 3.9. Detection of ADAM12 and MUC4 Isoforms at the Protein Level

The probable protein isoforms of ADAM12 and MUC4 reported in Uniprot were used as references for Western blotting (Appendix A). We attempted to detect the secreted ADAM12 protein in whole-cell extracts from all three cell lines. Using one anti-ADAM12 antibody (cat# orb373867), we identified a ~100 KDa isoform of ADAM12 that matched the expected size of Isoform 1 in the Uniprot database. The protein levels of this isoform were detected at the highest levels in SW480 cells, followed by SW620 cells, with the lowest levels detected in CCD841 cells (Appendix A). Another anti-ADAM12 antibody (cat# orb155592) detected the three remaining isoforms at similar levels in SW480 and SW620 cells but did not detect any protein in CCD841 cells (Appendix A).

Two different MUC4 antibodies detected isoform 1 at the expected size of 230 KDa. One of these antibodies (Cat# orb306041) could not detect any band in CCD841 cells (Appendix A), whereas the other antibody (Cat# orb399150) detected a ~232 KDa protein band (Appendix A). We attempted to identify secreted isoforms but could not detect either ADAM12 or MUC4 in conditioned media. This information was used to analyze the Western blots of ADAM12 and MUC4 in whole cell lysates from CCD841, SW480 and SW620 cells.

## 4. Discussion

In our previous study of tumor samples from patients with CRC, we identified novel genes that are differentially expressed as well as alternatively spliced in a highly significant manner, as expressed by their alternatively spliced scores. In the present study, we performed the first characterization of the expression of various isoforms of two of these alternatively spliced genes—*ADAM12* and *MUC4*—in a metastatic cell line model. Our study supports previous efforts to analyze spliceosome genes in 9070 patients across 27 types of cancer within the context of 3PM medicine [18]. We have provided a detailed analysis of pre-selected genes that will develop the field of alternative splicing for biomarker discovery to help in establishing the concept of 3P medicine and making it clinically viable.

Alternative splicing is a complex mechanism that results in multiple functional isoforms of the same gene and allows the genome to generate a diverse transcriptome and proteome [29]. It allows certain exons to be included or excluded from the final, mature mRNA, resulting in different protein isoforms and functional diversity. Approximately 95% of human genes with multiple exons undergo alternatively splicing during pre-mRNA maturation [30,31]. Changes in alternative splicing mechanisms leading to altered ratios of splice variants have been associated with cancer progression and metastasis [32], as well as specific stages of cancer. For example, the expression of the isoforms *CXCL12* and *IG20/MADD* changes during the progression of glioblastoma [33,34]. Several genes have been identified to be alternatively spliced in colorectal cancer [35]. Moreover, TAK1 exon 12 skipping has been shown to favor epithelial mesenchymal transition (EMT) and drug resistance in cancer cells [31]. Hallmarks of alternative splicing in cancer have been identified, surmounting the challenge of data availability and suggesting the inhibitors of alternatively spliced isoforms as anticancer therapeutic agents [36]. This suggestion may be debated in light of the evidence that alternatively splicing generates isoforms that are pro-apoptotic, but alternatively spliced isoforms can be a goldmine for biomarker discovery without any contest. A recent review noted that more than 15,000 alternative splicing events have been associated with cancer cell responsiveness to chemotherapy, cancer cell proliferation, invasion and resistance to apoptosis [37]. A comprehensive study of RNA-Seq data and corresponding clinical information showed that AS events are implicated in significant CRC-related processes, such as protein kinase activity and the PI3K-Akt and p53 signaling pathways. In addition, several alternative splicing events were connected to the OS and disease-free survival (DFS) of CRC patients, which suggests that alternative splicing events may affect the disease prognosis [34]. Our study further demonstrates the use of alternatively spliced genes as potential prognostic biomarkers, as exhibited by the survival curves generated using TCGA data.

Apart from understanding the functional role of these alternatively spliced isoforms in disease, alternatively spliced genes may also potentially represent biomarkers. Specific splice variants have been reported in several types of cancer [38,39,40]. Several genes were previously reported to be alternatively spliced in CRC [35]. The tumor location is an established prognostic indicator in CRC, and a recent study suggested that alternatively spliced genes were differentially expressed in left- and right-sided CRC [41].

*ADAM12*, a member of the disintegrin-containing metalloprotease family, has been reported to be involved in a variety of diseases, including cancer (Nyren-Erickson, 2013 #32). *ADAM12* exhibits diverse functions in both normal and pathological states, including the remodeling of the cell surface, the shedding of ectodomain and the regulation of the availability of growth factors and the interactions between cells and the extracellular matrix [42,43,44]. Several studies suggest that *ADAM12* plays a key role in the remodeling of the extracellular matrix, which is an important hallmark of neoplastic disease (Park, 2021 #33; Zhu, 2022 #34).

The roles of mucins in the initiation and progression of CRC are well characterized. The genes of the mucin family were found to be downregulated in the same patient cohort that we reported in this study. *MUC4* is alternatively spliced to generate secreted and membrane-associated proteins [45]. The genomic organization of *MUC4* is not well understood, and its transcriptional and post-transcriptional regulation in different cells and under various conditions is poorly characterized. RT-PCR-based studies showed that the alternative splicing of *MUC4* leads to the expression of secreted and membrane-associated proteins in various types of tissues [46]. This observation supports the results of our study, in which we demonstrated that the expression of *MUC4* is higher in CRC cell lines compared to that in CCD841 (a non-cancerous cell line). *MUC4* expression was higher in the metastatic SW620 than it was in the primary cell line SW480. On the other hand, the expression of the *MUC4-16* isoform is lower in the CRC cell line compared to that in CCD841. The *MUC4-16* isoform was higher in the primary cell line SW480 than it was in the metastatic SW620. These variations in the expression of *MUC4* isoforms suggest that *MUC4* might have different roles in cells, representing different cancer stages. Thus, the alternatively spliced isoforms could be used as a biomarker for colorectal cancer metastasis. Our results depicting the effect of 5-FU on the expression pattern of different isoforms suggest the probable use of isoforms as a predictive biomarker as well. Previous studies have taken a similar approach to make personalized medicine a reality and enable 3P medicine [47].

The evidence in favor of multi-omics is increasingly supporting the idea that will accelerate the practice of 3PM [25]. We took this approach and tried studying the protein isoforms of the *MUC4* and *ADAM12* genes, with an aim to establish secreted isoforms as biomarkers. We detected the membrane-bound *MUC4* protein but could not detect the secreted form. There has been no previous analysis of *MUC4* in any cell line. More sophisticated techniques are needed to detect this heavily glycosylated protein. Our results confirm the exon skipping reported in prior studies, as *MUC4* was one of the six genes that exhibited cancer tissue-specific differential exon skipping [45,48]. The 223 bp *MUC4-16* amplicon contains a single nucleotide difference compared to *MUC4-1* and *MUC4-6*. *MUC4-1* has a C nucleotide at position 1934, whereas *MUC4-6* has an adenine (A). We were able to detect the *MUC4-6* (with the ‘A’ nucleotide) using Sanger sequencing. There is no annotation for *MUC4-6* in ENSEMBL. However, *MUC4-1* is annotated by ENSEMBL, and RNA-Seq based on ENSEMBL confirmed the expression of *MUC4-1*. *MUC4-1* was expressed at lower levels in comparison to other isoforms, and this may explain why we were not able to detect *MUC4-1* by Sanger sequencing. *MUC4-6* may be expressed at much higher levels than *MUC4-1*; thus, after amplification, Sanger sequencing was only able to detect *MUC4-6*.

This study provides evidence to further suggest that alternatively spliced isoforms hold potential as biomarkers for CRC that can aid in the practice of 3PM. Comprehensive analyses of gene organization will help to advance our knowledge of the largely unstudied mechanisms of alternative splicing. Using prior evidence, we selected genes that were highly likely to represent candidate biomarkers for CRC. Based on this study, we conclude that the *ADAM12* and *MUC4* isoforms may be potential candidate biomarkers for metastasis in CRC. However, more work is needed to detect the secreted forms of *ADAM12* and *MUC4*. The further design of primers that can exclusively amplify a single isoform is necessary to study the clinical value of these isoforms as prognostic biomarkers. A limitation of this study was the inability to correlate the gene isoforms with the identified proteins.

## 5. Conclusions

This study reveals that the expression of the isoforms of two genes, *ADAM12* and *MUC4,* varies in a cell line model of CRC metastasis, and 5-FU drug treatment altered the expression patterns of these isoforms. Variations in these isoforms between cell lines were also observed at the protein level using different antibodies. This evidence supports the potential of alternatively spliced isoforms as prognostic and predictive biomarkers for cancer, which will be critical in the implementation of 3PM.

## Figures and Tables

**Figure 1 jpm-13-00135-f001:**
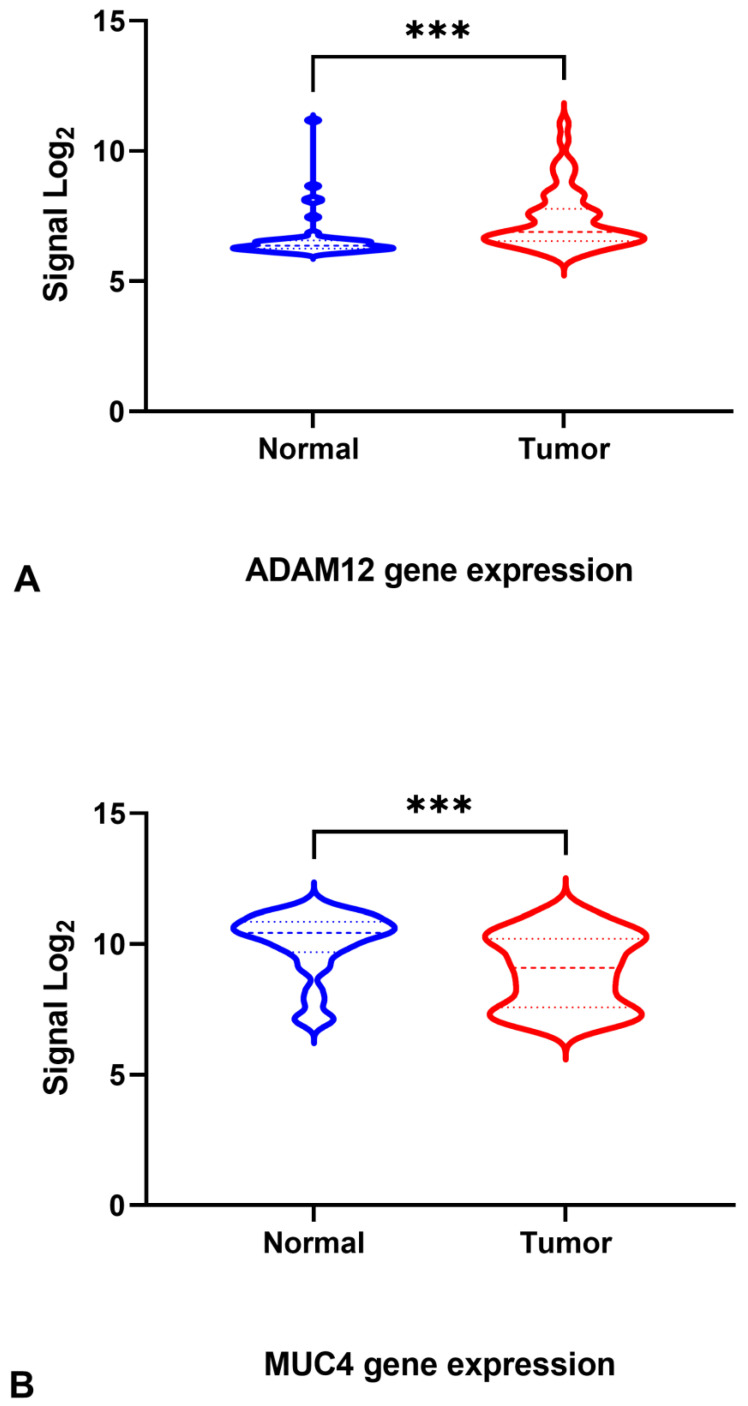
**Analysis of microarray data for tumor and adjacent normal samples from patients with CRC.** (**A**): Expression of *ADAM12* in a previous microarray study. *ADAM12* was significantly upregulated in tumor samples. (**B**): The expression of *MUC4* in a previous microarray study. *MUC4* was significantly downregulated in tumor samples. Graphs represent data from 48 patients with CRC. Welch’s *t*-test was used to determine statistical significance. *** denotes *p*-value < 0.001.

**Figure 2 jpm-13-00135-f002:**
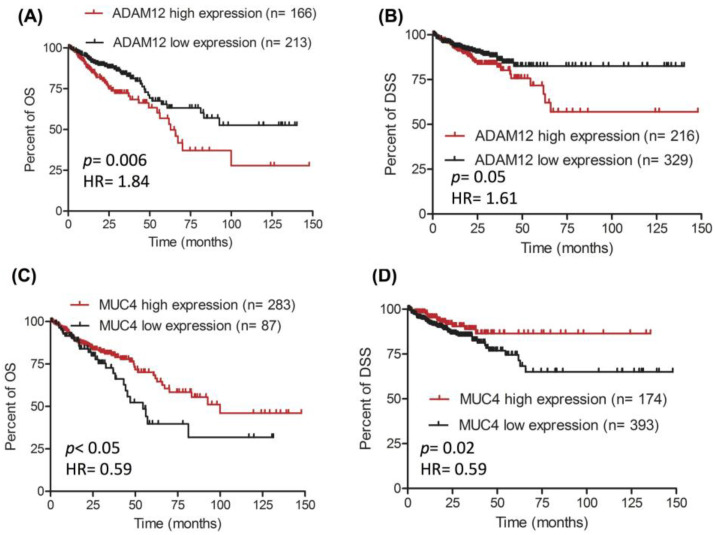
**Survival curves of the prognostic value of *ADAM12* and *MUC4* in colorectal adenocarcinoma.** Two TCGA transcriptomics datasets generated from patients with colorectal adenocarcinoma were used to evaluate the prognostic potential of *ADAM12* and *MUC4*. A high expression of *ADAM12* predicted a shorter OS (**A**) colorectal adenocarcinoma TCGA Firehose Legacy dataset) and a shorter DSS (**B**) colorectal adenocarcinoma TCGA PanCancer Atlas dataset). A low expression of *MUC4* predicted a shorter OS (**C**) colorectal adenocarcinoma TCGA Firehose Legacy dataset) and a shorter DSS (**D**) colorectal adenocarcinoma TCGA PanCancer Atlas dataset). OS: overall survival; DSS: disease-specific survival.

**Figure 3 jpm-13-00135-f003:**
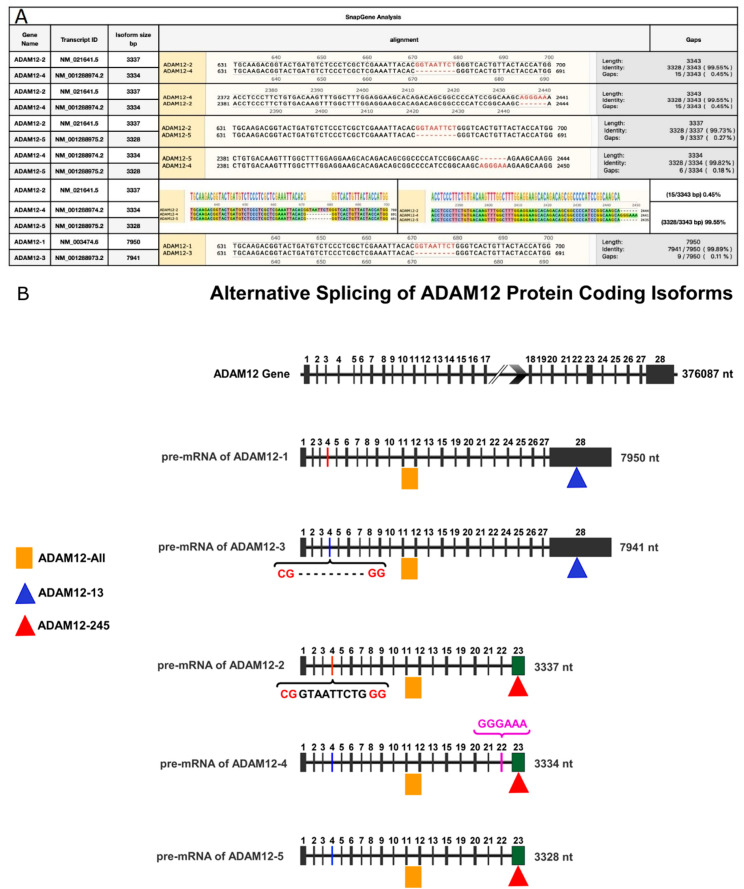
**Sequence similarity analyses for *ADAM12*.** (**A**): Sequence similarity of the five alternatively spliced isoforms of *ADAM12*. (**B**): Illustration of the exon organization of *ADAM12* used to design primers for the specific amplification of alternatively spliced isoforms. The *ADAM12* gene and isoforms are presented as horizontal lines. Each vertical bar represents an exon separated by intronic regions. The image shows the skipping of exon 14 in all isoforms and the pretermination of isoforms 2, 4 and 5.

**Figure 4 jpm-13-00135-f004:**
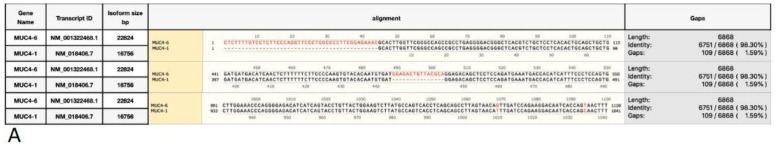
**Sequence similarity analyses for *MUC4*.** (**A**): The sequence similarity of the four alternatively spliced isoforms of *MUC4*. (**B**): Illustration of the exon organization of *MUC4* used to design primers for the specific amplification of alternatively spliced isoforms. The *MUC4* gene and isoforms are presented as horizontal lines. Each vertical bar represents an exon separated by intronic regions. For *MUC4-6*, we used the first 44 bp in exon 1 that are unique and the slight difference in exon 3 as the basis for the primer design.

**Figure 5 jpm-13-00135-f005:**
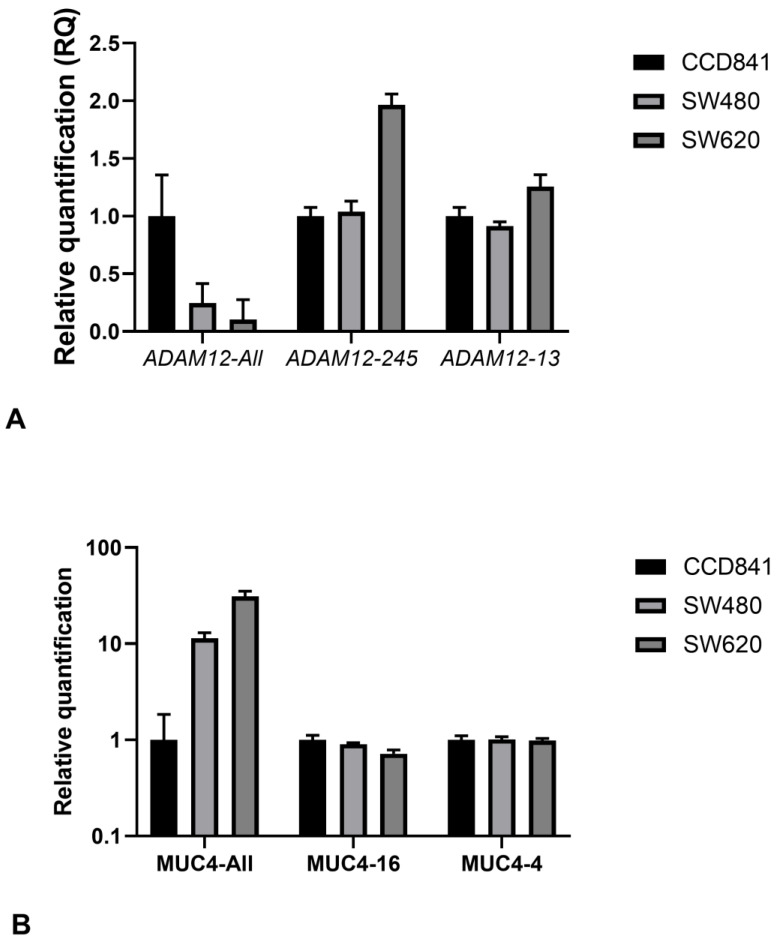
**Quantification of the expression of *ADAM12* and *MUC4* isoforms in cell lines.** Quantitative RT-PCR analyses indicated the differential expression of the isoforms of *ADAM12* and *MUC4* in normal colon (CCD841) cells and primary (SW480) and metastatic (SW620) CRC cell lines. (**A**) The trend for *ADAM12*-All amplicon expression was CCD841 > SW480 > SW620, whereas the trend for *ADAM12-245* amplicon expression was CCD841 = SW480 > SW620. A similar trend was observed for the *ADAM12-13* primers. (**B**) *MUC4*-All showed an increasing pattern from normal to metastatic cells, whereas no differences in the *MUC4-16* and *MUC4-4* amplicons were observed between the cell lines.

**Figure 6 jpm-13-00135-f006:**
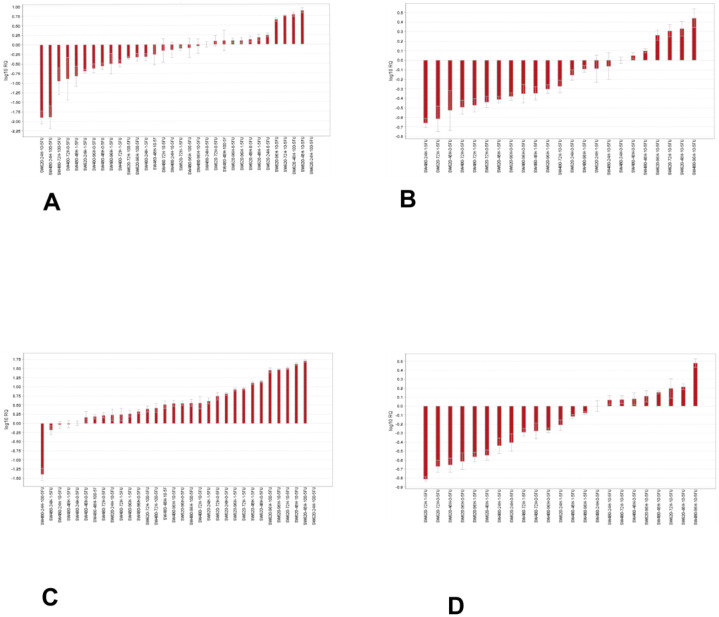
Effect of 5-FU on the expression of *ADAM12* and *MUC4* isoforms. Waterfall plots for SW480 and SW620 cells treated with 0, 1, 10 or 100 *µ*M 5-FU for 24, 48, 72 or 96 h. The expression of isoforms detected by the primers for (**A**) *ADAM12*-All, (**B**) *ADAM12-245*, (**C**) *MUC4*-All and (**D**) *MUC4-4*.

**Table 1 jpm-13-00135-t001:** Primer pairs used to amplify the alternatively spliced protein coding isoforms of *MUC4* and the housekeeping genes.

Gene Isoform	Transcript ID	Primer Sequence	Amplicon Size (bp)
*ADAM12*- All		F-GTAGCTGTCAAATGGCG R-CCACAAATCTGTTCCCAC	198 bp
*ADAM12-1*	ENST00000368679.4 * NM_003474.6 **	F-CCAACTCGTATAGCATGCATC	197 bp
*ADAM12-3*	ENST00000448723.1 * NM_001288973.2 **	R-CAATGCCCACGTAATGCAC	
*ADAM12-2*	ENST00000368676.4 * NM_021641.5 **	F-CTGCTCCTGAGAGAGTAGC	
*ADAM12-4*	NM_001288974.2 **		205 bp
*ADAM12-5*	NM_001288975.2 **	R-CAGAGCATTAAGTTGCAGCC	
*ADAM12-4*	NM_001288974.2 **	F-GAAAGAAGCAAGGCAGGA R-CGTTTCCATGACAACAGAC	231 bp
GAPDH	ENSG00000111640 * NG_007073.2 **	F-ACCCAGAAGACTGTG R-CAGTGAGCTTCCCGTTCAG	139 bp
ACTNB	ENSG00000075624 * NG_007992.1 **	F-TGACGTGGACATCCGCAAAG R-CTGGAAGGTGGACAGCGAGG	205 bp

* ENSEMBL database, ** NCBI database.

**Table 2 jpm-13-00135-t002:** FPKM values for *ADAM12* and *MUC4* transcripts obtained from RNA-Seq data.

Transcript ID	SW620	SW480	Fold Change
			(SW480/SW620)
ADAM12_ENSG00000148848.10	0.0245429	0.0241492	0.983958701
ADAM12_ENST00000368679.4	0.00916641	0.000339521	0.037039692
ADAM12_ENST00000368676.4	0.0273946	0.0424578	1.549860191
ADAM12_ENST00000467145.1	0	5.45E-05	
ADAM12_ENST00000482291.1	0.0861337	9.29E-05	0.001079091
ADAM12_ENST00000485388.2	0.010143	0	0
ADAM12_ENST00000448723.1	0.0269748	0.00363765	0.134853641
ADAM12_ENST00000494661.1	0	1.44E-18	
MUC4_ENST00000415455.1	4.40E-06	4.54E-10	1.03E-04
MUC4_ENST00000308466.8	7.57E-09	1.71E-12	2.26E-04
MUC4_ENST00000392407.2	0.112782	0.0177344	1.57E-01
MUC4_ENST00000339251.5	8.73E-06	0.0596096	6.83E+03
MUC4_ENST00000448861.1	0.00120339	0.00127307	1.06E+00
MUC4_ENST00000349607.4	2.30E-07	4.93E-13	2.14E-06
MUC4_ENST00000346145.4	6.92E-12	2.68E-07	3.87E+04
MUC4_ENST00000478156.1	1.28E-91	1.11E-25	8.67E+65
MUC4_ENSG00000145113.17	0.0510692	0.0332947	6.52E-01
MUC4_ENST00000463781.3	1.02E-66	5.30E-23	5.20E+43
MUC4_ENST00000479406.1	2.42E-93	6.33E-30	2.62E+63
MUC4_ENST00000462323.1	3.35E-94	2.96E-32	8.84E+61
MUC4_ENST00000475231.1	9.90E-95	1.02E-32	1.03E+62
MUC4_ENST00000470451.1	5.26E-94	4.40E-32	8.37E+61
MUC4_ENST00000480843.1	1.55E-94	1.51E-32	9.74E+61
MUC4_ENST00000477086.1	3.49E-90	2.11E-30	6.05E+59
MUC4_ENST00000466475.1	8.35E-91	6.02E-31	7.21E+59
MUC4_ENST00000477756.1	1.63E-90	1.08E-30	6.63E+59
MUC4_ENST00000464234.1	1.63E-56	1.69E-12	1.04E+44
MUC4_ENST00000467235.1	0.0118175	0.0397415	3.36E+00
MUC4_ENST00000469992.1	2.94E-76	0.153917	5.24E+74
MUC4_ENST00000486425.1	2.02E-234	6.08E-94	3.01E+140
MUC4_ENST00000478685.1	2.08E-76	0.11691	5.62E+74

**Table 3 jpm-13-00135-t003:** RNA-Seq analysis of fold changes in splicing factor genes between metastatic (SW620) and primary (SW480) colorectal cancer cell lines. (Values are normalized FPKM values from RNA-Seq analyses.)

Gene	SW620	SW480	Fold Change (SW480/SW620)
SF3B1	287.3	242.6	1.18
U2Af1	91.08	109.47	0.83
U2AF1L4	1.228	2.64	0.47
SRSF2	225.8	200.5	1.12
RBM10	124.05	89.05	1.39
FUBP1	157.3	104.5	1.5
SRPK1	123.48	118.4	1.04

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
