# Peer review of "Alternatively Spliced Isoforms of MUC4 and ADAM12 as Biomarkers for Colorectal Cancer Metastasis"

_jpm, 2023, doi:10.3390/jpm13010135_

Round 1

Reviewer 1 Report

The Article Titled "Alternatively spliced isoforms of MUC4 and ADAM12 as biomarkers for colorectal cancer metastasis", authored by Saleh Althenayyan and colleagues is a state-of-the-art study on the development of novel biomarkers. I would like to congratulate the authors on their study design and the provided data to support their conclusions. 

I have the following minor comments:

1) I would recommend that the authors decide to present some of their data as supplementary, since there are several figures and tables in the manuscript. My suggestion would be Table 2, Figure 5, Figure 6, Figure 9 and Table 5.

2) There are few errors throughout the text that the authors should consider revising. In addition line numbers were in the text, that was making the reading difficult in some instances. However, I believe that issue is based on the line numbering from the Journal and may not impact the final text.

Author Response

  1. I would recommend that the authors decide to present some of their data as supplementary, since there are several figures and tables in the manuscript. My suggestion would be Table 2, Figure 5, Figure 6, Figure 9 and Table 5.

Response from authors: We thank the reviewer for this comment. We agree with the reviewer and therefore attached a new file of supplementary figures and tables. The text was adjusted accordingly.

  1. There are few errors throughout the text that the authors should consider revising. In addition line numbers were in the text, that was making the reading difficult in some instances. However, I believe that issue is based on the line numbering from the Journal and may not impact the final text.

Response from authors: We are very thankful to the reviewer for pointing this out. We have thoroughly checked the manuscript, made necessary changes and updated. Line numbers are provided by the submission software of mdpi publishers. We will provide them the feedback.

Reviewer 2 Report

Dear Authors,

This is an impressive study describing the potential of alternatively spliced isoforms as predictive biomarkers in PPPM approach towards cancer. Following are the few suggestions to improve the readability of manuscript.

Major Comments

1.     Please explain why did the high expression of ADAM12 have poorer overall survival [OS] but better disease specific survival [DSS] events of 16% compared to 11% in the low expression group on Line 219?

2.     Please explain why did the high expression of MUC4 have better overall survival [OS] but poorer disease specific survival [DSS] events of 8% compared to 15% in the low expression group on Line 226?

3.     The details given on line 305, “However, the MUC4-16 primers detected- - ” doesn’t match with information on Figure 7B. Please correct.

4.     What statistical test was used to determine the significance of differences in quantification of expression of ADAM12 and MUC4 isoforms on Figure 7.

5.     Figure 8 is too small to visualize the waterfall plots. Please add high resolution figures.

6.     Under section 3.7, what was the criteria to select different 5-FU dose levels. Please elaborate on the effect of 5-FU treatment on ADAM12 and MUC4 isoforms expression in primary and metastatic CRC cell lines. What is the possible explanation for higher dose levels of 5-FU treatment resulting into higher expression of both ADAM12 and MUC4 isoforms.

Minor Comments

1.     Line numbers are disturbed throughout the manuscript. Please use “Line Numbers” option under layout tab to correct the issue.

2.     For language consistency either use B-ACTIN or ACTB on Figure 6 and its caption.

3.     For language consistency either use AS or ‘alternatively spliced’ isoforms throughout the text.

4.     On abstract please give 5-FU in parenthesis at its first mention on Line 14.

5.     Please give full form of DFS on Line 385.

6.     Please add references for Line 395 and Line 399.

7.     On Figure 2, caption, please mention what does HR mean.

8.     On Figure 6, please correct “The band intensity of ADAM12-All” to MUC4 -All.

9.     On Figure 7. Please correct the wording “A similar trend was observed for ADAM12-13” as it appeared to be similar for all cell lines for this primer.

10.  On Figure 8, please correct the caption from “Effect of 5-FU of” to “Effect of 5-FU on”.

11.  On Line 33, sub-group number iv) is missing.

12.  On Line 68, there is an extra symbol ). Numbering could be removed on this statement.

13.  On Line 121 and 122, please correct for the symbol ºC.

14.  On Line 132, does it mean Nanodrop Spectrophotometers?

15.  Inverted commas used for “National Cancer- - -” should be removed from Line 193 and 194.

hanks & Regards,

Author Response

Reviewer 2

Major Comments

  1. Please explain why did the high expression of ADAM12 have poorer overall Survival [OS] but better disease-specific Survival [DSS] events of 16% compared to 11% in the low expression group on Line 219?

Response from authors: We thank the reviewer for this comment. Both the text and the figure (Fig 2A and 2B) show that high expression of ADAM12 is associated with poorer OS and DSS. This proposed ADAM12 as a poor prognostic marker for OS and DSS. A text was added to further explain this point.

  1. Please explain why did the high expression of MUC4 have better overall Survival [OS] but poorer disease-specific Survival [DSS] events of 8% compared to 15% in the low expression group on Line 226?

Response from authors: We thank the reviewer for this comment. Both the text and the figure (Fig 2C and 2D) show that low-expression of MUC4 is associated with poorer OS and poorer DSS. This proposed MUC4 as a poor prognostic marker for OS and DSS. A text was added to further explain this point.

  1. The details given on line 305, “However, the MUC4-16 primers detected- - ” doesn’t match with information on Figure 7B. Please correct.

Response from authors: We thank the reviewer for this comment. The text has been matched with the figure to “However, the MUC4-16 primers detected the highest levels of transcripts in CCD841 cells, lower levels in SW480 cells and lowest  levels in SW620 cells”

  1. What statistical test was used to determine the significance of differences in quantification of expression of ADAM12 and MUC4 isoforms on Figure 7.

Response from authors: Prism software (GraphPad) and Microsoft Excel were employed to analyze the data and calculate p-values using the statistical tests indicated within the figure legends. Differences between groups were considered significant for p values ≤ 0.05.  Figure 1 depicting patient comparisons used Welch’s t-test. Figure 5 &6 used unpaired t-tests.

  1. Figure 8 is too small to visualize the waterfall plots. Please add high resolution figures.

Response from authors: We have redrawn the figure. It is numbered as Figure 6 due to rearrangement of figures as suggested by another reviewer.

  1. Under section 3.7, what were the criteria to select different 5-FU dose levels. Please elaborate on the effect of 5-FU treatment on ADAM12 and MUC4 isoforms expression in primary and metastatic CRC cell lines. What is the possible explanation for higher dose levels of 5-FU treatment resulting into higher expression of both ADAM12 and MUC4 isoforms.

Response from authors: We used three doses varying on logarithmic scale (1,10, &100 µM). These doses are close to physiological doses used in the clinic. There was a varied response to 5-FU in terms of dosage and duration by both primary (SW480) and metastatic cells (SW620). Our objective was to assess the response of MUC4 and ADAM12 expression to dose and duration which is clearly exhibited in the waterfall plot. Further studies may delineate the mechanism of these expression changes. Since 5-FU works through the inhibition of thymidylate synthase, there are several pathways that may affect the gene expression directly. Change in expression in both MUC4 and ADAM12 would suggest a general effect on transcription rather than being specific.

Minor Comments

  1. Line numbers are disturbed throughout the manuscript. Please use "Line Numbers" option under the layout tab to correct the issue.

Response from authors: We thank the reviewer for this comment. The issue with line numbering is due to the line numbering system of the Journal, as there is no numbering issue in the originally submitted form.

  1. For language consistency either use B-ACTIN or ACTB on Figure 6 and its caption.

Response from authors: We thank the reviewer for this comment. It has been corrected to B-ACTIN and  moved to the supplementary file

  1. For language consistency either use AS or ‘alternatively spliced’ isoforms throughout the text.

Response from authors: We thank the reviewer for this comment. We unified the language to alternatively spliced.

  1. On abstract please give 5-FU in parenthesis at its first mention on Line 14.

Response from authors: We thank the reviewer for this comment. We edited the abstract and added (5-FU)

  1. Please give a full form of DFS on Line 385.

Response from authors: We thank the reviewer for this comment. We edited DFS to disease-Free Survival

  1. Please add references for Line 395 and Line 399.

Response from authors: We thank the reviewer for this comment. We have added the required references.

  1. On Figure 2, caption, please mention what does HR mean.

Response from authors: HR: hazard ratio of high-expression group versus low-expression group. This was added to the figure caption.

  1. On Figure 6, please correct “The band intensity of ADAM12-All” to MUC4 -All.

Response from authors: We thank the reviewer for this comment. We edited the issue and the figure was moved to (Supplementary Figure 3)

  1. On Figure 7. Please correct the wording “A similar trend was observed for ADAM12-13” as it appeared to be similar for all cell lines for this primer.

Response from authors: We thank the reviewer for this comment. I respectfully disagree with the reviewer's comment that ADAM12-13 upregulated in SW620 matches the trend of ADAM12-245. Also, the new figure number is Figure 5 as we moved some of the figures to supplementary.

  1. On Figure 8, please correct the caption from “Effect of 5-FU of” to “Effect of 5-FU on”.

Response from authors: We thank the reviewer for this comment. The word was corrected

  1. On Line 33, sub-group number iv) is missing.

Response from authors: We thank the reviewer for this comment. ( iv) was added

  1. On Line 68, there is an extra symbol ). Numbering could be removed on this statement.

Response from authors: We thank the reviewer for this comment. The numbering was removed and the extra symbol.

  1. On Line 121 and 122, please correct for the symbol oC.

Response from authors: We thank the reviewer for this comment. The symbols were correct.

  1. On Line 132, does it mean Nanodrop Spectrophotometers?

Response from authors: We thank the reviewer for this comment. The word Spectrophotometer was added.

  1. Inverted commas used for “National Cancer- - -” should be removed from Line 193 and 194.

Response from authors: We thank the reviewer for this comment. The inverted commas were removed.

Round 2

Reviewer 1 Report

The authors have addressed my comments. I recommend acceptance 

Author Response

We would like to thank the reviewer for taking out time to read the manuscript and recommend acceptance.

Reviewer 2 Report

Alternatively spliced isoforms of MUC4 and ADAM12 as biomarkers for colorectal cancer metastasis

Please see comments below:

1.     I do not see added references for Line 395 (ADAM12, a member of the - - of diseases, including cancer.) and Line 399 (Several studies suggest ADAM12 plays a key role--).

2.     I do not see mention of unpaired t-tests on the captions for Figure 5 &6.

3.     On Figure 5B, almost matching shades create confusion in black-white print. Use of different pattern fill can clearly differentiate between (SW480) and metastatic (SW620) CRC cell lines.

4.     Information given in text doesn’t match Figure 5 caption. Please see below in yellow. Also, for MUC4-4, the expression levels might be higher, but the difference is almost negligible (no statistical significance). Please see below in blue.

As per text (Line 305 to 309)

However, the MUC4-16 primers detected the highest levels of transcripts in CCD841 cells, lower levels in SW480 cells and lowest levels in SW620 cells. The MUC4-4 primers resulted in a similar expression pattern as the MUC4-16 primers, but with larger differences in the MUC4-4 expression values between the three cell lines (Figure 5B).

Figure 5. Quantitative RT-PCR analyses indicated differential expression of the isoforms of ADAM12 and MUC4 in normal colon (CCD841) cells and primary (SW480) and metastatic (SW620) CRC cell lines. A: The trend for ADAM12-All amplicon expression was CCD841 > SW480 > SW620, whereas the trend for ADAM12-245 amplicon expression was CCD841 = SW480 > SW620. A similar trend was observed for the ADAM12-13 primers. B: MUC4-All showed an increasing pattern from normal to metastatic cells, whereas no differences in the MUC4-16 and MUC4-4 amplicons were observed between the cell lines.

Thanks & Regards,

Author Response

We would like to thank the reviewer for taking out time to read the manuscript and suggest changes that will improve the manuscript. Please find below the response to reviewer's comments:

  1. I do not see added references for Line 395 (ADAM12, a member of the - - of diseases, including cancer.) and Line 399 (Several studies suggest ADAM12 plays a key role--).

Response from the authors: We thank the reviewer for this comment. We agree with the reviewer, and we added the proper references for each sentence. The endnote file of these references are also submitted separately. Three references were added.

  1. I do not see mention of unpaired t-tests on the captions for Figure 5 &6.

Response from the authors: We thank the reviewer for this comment. We agree with the reviewer, and we added the following sentences to Figures 5 and 6 "Unpaired t-test was used to determine statistical significance."

  1. On Figure 5B, almost matching shades create confusion in black-white print. Use of different pattern fill can clearly differentiate between (SW480) and metastatic (SW620) CRC cell lines.

Response from the authors: We would like to thank the reviewer for suggesting this improvement. We will definitely request the publishers to incorporate these changes in the final print.

  1. Information given in text doesn’t match Figure 5 caption. Please see below in yellow. Also, for MUC4-4, the expression levels might be higher, but the difference is almost negligible (no statistical significance). Please see below in blue.

Response from the authors: We thank the reviewer for this comment. We agree with the reviewer's comments, and the sentence has been changed in the figure caption to "B: MUC4-All showed an increasing pattern from normal to metastatic cells, whereas MUC4-16 and MUC4-4 amplicons were observed to downregulate from normal to metastatic cell lines CCD841 > SW480 > SW620. Unpaired t-test was used to determine statistical significance". Also, the text was changed to "The MUC4-4 primers resulted in a similar expression pattern as the MUC4-16 primers, by a decrease in expression from normal to metastatic cell lines, but no significant difference in expression values between the three cell lines (Figure5B)"